# Survey of Recent Deep Neural Networks with Strong Annotated Supervision in Histopathology

Dominika Petríková [1,2,*] and Ivan Cimrák [1,2]

1    Cell-in-Fluid Biomedical Modelling & Computations Group, Faculty of Management Science and Informatics, University of Žilina, Univerzitná 8215/1, 010 26 Žilina, Slovakia
2    Research Centre, University of Žilina, Univerzitná 8215/1, 010 26 Žilina, Slovakia
*    Correspondence: dominika.petrikova@fri.uniza.sk

**Abstract:** Deep learning (DL) and convolutional neural networks (CNNs) have achieved state-of-the-art performance in many medical image analysis tasks. Histopathological images contain valuable information that can be used to diagnose diseases and create treatment plans. Therefore, the application of DL for the classification of histological images is a rapidly expanding field of research. The popularity of CNNs has led to a rapid growth in the number of works related to CNNs in histopathology. This paper aims to provide a clear overview for better navigation. In this paper, recent DL-based classification studies in histopathology using strongly annotated data have been reviewed. All the works have been categorized from two points of view. First, the studies have been categorized into three groups according to the training approach and model construction: 1. fine-tuning of pre-trained networks for one-stage classification, 2. training networks from scratch for one-stage classification, and 3. multi-stage classification. Second, the papers summarized in this study cover a wide range of applications (e.g., breast, lung, colon, brain, kidney). To help navigate through the studies, the classification of reviewed works into tissue classification, tissue grading, and biomarker identification was used.

**Keywords:** classification; convolutional neural networks; deep learning; digital pathology; histology image analysis

## 1. Introduction

Traditionally, pathology diagnosis has been performed by a human pathologist observing stained specimens from tumors on glass slides using a microscope to diagnose cancer. In recent years, deep learning has rapidly developed, and more and more entire tissue slides are being captured digitally by scanners and saved as whole slide images (WSIs) [1]. Since a large amount of WSIs are being digitized, it is only natural that many attempts have been made to explore the potential of deep learning on histopathological image analysis. Histological images and tasks have unique characteristics, and specific processing techniques are often required [2]. The authors in [3] carried out an extensive and comprehensive overview of deep neural network models developed in the context of computational histopathology image analysis. Their survey covers the period up to December 2019. Since the volume of research in this domain is rapidly growing, the aim of this review is to complement their overview with papers published since 2020. In contrast to their survey, the focus of this review is on a specific area of supervised learning only, namely classification using strongly annotated data.

The rest of this paper is organized as follows. In Section 2, a basic overview of neural networks used in the context of computational histopathology is presented. Section 3 discusses in detail supervised deep learning models and approaches used in digital pathology for classification tasks. These approaches have been grouped into three main categories: one-stage classification using fine-tuning, one-stage classification training models from

scratch, and the multi-stage classification approach. In Section 4, we discuss the histopathological point of view by classifying the methods according to their area of application. In Section 5, we conclude the paper.

## 2. Materials and Methods—Convolutional Neural Network

For this survey, only papers that performed classification of histological images with common convolutional neural network models and used strongly annotated datasets were selected. Other articles that used more complex deep learning models or weak annotations were not included in this review. The review was carried out by searching mostly through PubMed and also arXiv for articles containing deep learning (DL) keywords such as "convolutional neural networks", "classification", "deep learning", and histology keywords such as "hematoxylin and eosin", "H&E", and "histopathology" in the title or abstract. To narrow down the selection, combinations of deep learning keywords with histology keywords were used, for example, "CNN hematoxylin and eosin". The combination "deep learning histopathology" was omitted since both words are too general. Moreover, only articles published since 2020 have been searched. The subsequent filtering process can be described in four steps. The first two steps were designed to quickly filter out articles that were obviously irrelevant to the topic of this review and thus reduce as much as possible the number of articles that needed to be analyzed in more detail in the remaining two steps. In the first step, articles were filtered based on the title. Papers that were obviously not related to CNN's application for histological image data classification were excluded. This resulted in approximately 700 papers. Articles that could not be unambiguously excluded based on the title were filtered in a second step based on reading the abstract. In the third step, the introduction was analyzed. The main purpose was to exclude studies that did not meet the criteria of this review, such as papers using more complex deep learning approaches than convolutional neural networks or datasets not only consisting of histological images. In the last step, approximately 100 articles were fully read. This part was mainly focused on filtering out studies that only worked with strongly annotated datasets. We also included some papers that were missing from the initial search but were cross-referenced in selected articles.

The purpose of this chapter is to explain the concepts and models of deep neural networks (DNNs) used for classification tasks in digital pathology. Machine learning is a type of artificial intelligence that allows computers to learn and modify their behavior based on training data [4]. Supervised learning methods are the most commonly used, where the dataset consists of input features and corresponding labels. In the case of classification, the label represents one of a fixed number of classes. The algorithm learns patterns and connections in the data to find a suitable function that maps inputs to outputs, creating a model that captures hidden properties in the data and can be used to predict outputs for new inputs. Training a model involves finding the best model parameters that predict the data based on a defined loss function [5,6].

Neural networks are the foundation of most DNN algorithms, consisting of interconnected units called neurons organized into layers, including input, hidden, and output layers. DNNs have multiple hidden layers. A neuron's output, or activation, is a linear combination of its inputs and parameters (weights and bias) transformed by an activation function. Common activation functions in neural networks include sigmoid, hyperbolic tangent, and ReLU functions. At the final output layer, activations are mapped to a distribution over classes using the softmax function [6,7].

One of the most popular and commonly used supervised deep learning networks is CNNs, which are often employed for visual data processing of images and video sequences [8–10]. CNNs consist of three types of layers: convolutional layers, pooling layers, and fully connected layers, as shown in Figure 1. The convolutional layer is the most significant component of the CNN architecture. It consists of several filters, also called kernels, which are represented as a grid of discrete values. These values are referred to as kernel weights and are tuned during the training phase. The convolution operation

consists of the kernel sliding over the whole image horizontally and vertically. Additionally, the dot product is calculated between the image and kernel by multiplying corresponding values and summing up to create a scalar value at each position. In particular, each kernel is convolved over the input matrix to obtain a feature map. Subsequently, the feature maps generated by the convolutional operation are sub-sampled in the pooling layer. The convolution and pooling layers together form a pipeline called feature extraction. Above all, the fully connected layers combine the features extracted by the previous layers to perform the final classification task [8,11,12].

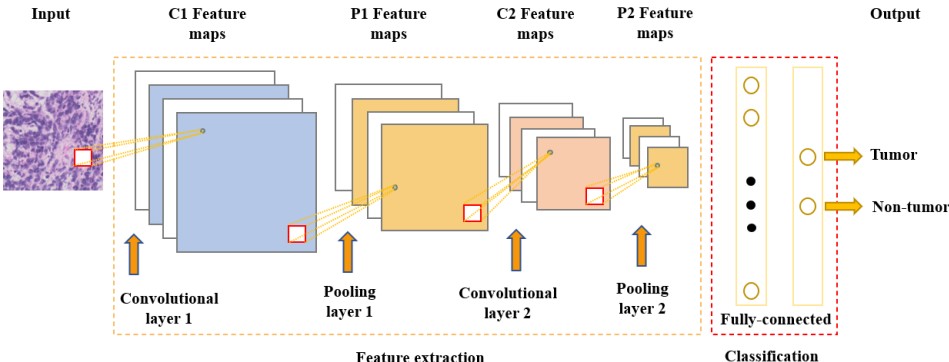

**Figure 1.** Convolutional neural network architecture.

## 3. Classification of Histopathology Images

This section provides a general overview of recent publications using deep learning and convolutional neural networks (CNNs) in digital pathology. The focus of this work is solely on supervised learning tasks applied for the classification of histological images. This category includes models that perform image-level classification, such as tumor subtype classification and grading, or use a sliding window approach to identify tissue types. Most deep learning approaches do not use the whole-slide image (WSI) as input because it would be computationally expensive (high dimensionality). Instead, they extract small square patches and assign a label to them. Existing methods can be grouped according to the level of annotations they employ. Based on the type of annotations used for training, two subcategories may be identified: the strong-annotations approach (patch-level annotations) and the weak-annotations approach (slide-level annotations) [13]. The first approach relies on the identification of regions of interest and the detailed localization of tumors by certified pathologists, while for the latter approach, it is sufficient to assign a specific class to a whole-slide image. In this work, a survey of the strong-annotations approach is conducted.

### 3.1. Strong-Annotations Approach (Patch-Level Annotation)

Referring to patch-level annotations as strong means that all extracted patches have their own label class. Typically, patch labels are derived from pixel-level annotations. Manually annotating pixels is very time-consuming and laborious work requiring an expert approach. For instance, pathologists have to localize and annotate all pixels or cells in WSI by contouring the whole tumor. This approach is shown in Figure 2. Therefore, there are currently very few strongly annotated histological images. Besides whole-slide image classification, pixel-wise/patch-wise predictions with the sliding window method enable spatial predictions such as localization and detection of cancerous cells/tissue. In addition, stacking patch predictions next to each other builds a WSI heatmap, so the model can be considered interpretable. Multiple examples of using CNNs in the problem of patch classification employ a single-stage approach when the patch is classified using one CNN architecture. In contrast, several approaches use a multi-stage workflow, where typically the output of one CNN architecture is fed into another CNN that delivers the final decision. Of course, even more CNN models can be included in such a workflow that can be labeled as multi-stage classification. For the one-stage approach, one can differentiate between

models that have been trained from scratch with artificially initiated weights and models that use pre-trained CNN architectures on data often not related to the original problem. For multi-stage problems, such differentiation becomes difficult due to many possibilities, since some CNNs from the multi-stage workflow may be trained from scratch, while others may be pre-trained. In Figure 3, the top graphic shows the categorization of CNN methods used in this section.

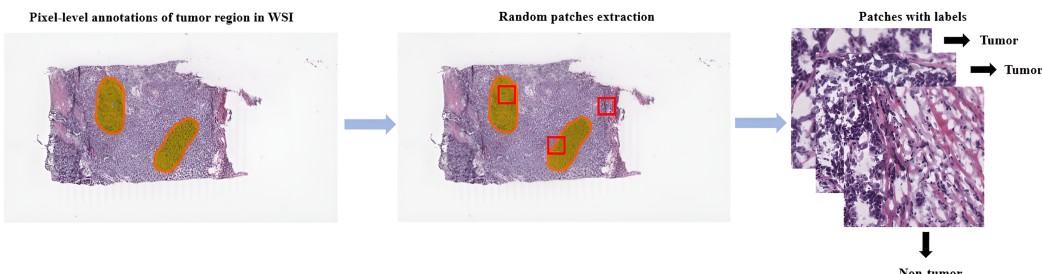

**Figure 2.** Construction of patches from pixel-level annotations of WSI.

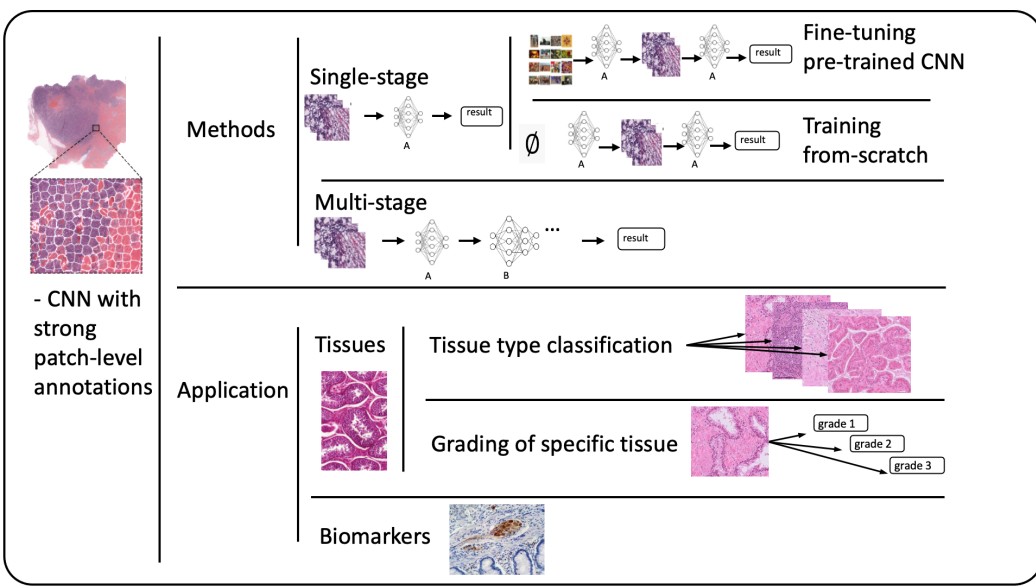

**Figure 3.** Methods: Categorization of CNN methods used in Section 3. Application: Categorization of application areas used in Section 4.

### 3.2. Fine-Tuning

The easiest way of training CNNs with a limited amount of data is using one of the well-known pre-trained architectures. Typically, models are initialized using weights pre-trained on ImageNet and fine-tuned on histopathological images. Papers using this approach are summarized in Table 1. In [14], the authors fine-tuned VGGNet [15], ResNet [16], and InceptionV4 [17] models to obtain the probabilities of small patches (100 × 100 pixels), being tumor-infiltrating lymphocyte (TIL)-positive or TIL-negative extracted from WSIs of 23 cancer types. For the region classification performance, they extracted bigger super-patches (800 × 800 pixels) and annotated them with three categories (Low TIL, Medium TIL, or High TIL) based on the ratio of TIL-positive area. To obtain a prediction of the category, super-patches were divided into an 8x8 grid and each square (100 × 100 pixel patch) was classified as TIL-positive or TIL-negative. Subsequently, the correlation between the score of CNN (number of positive patches in super-patch) and pathologists' annotations was observed. In [18], they developed a deep learning-based six-type classifier for the identification of a wider spectrum of lung lesions including lung cancer. Furthermore,

they also included pulmonary tuberculosis and organizing pneumonia, which often needs to be surgically inspected to be differentiated from cancer. EfficientNet [19] and ResNet were employed to carry out patch-level classification. To aggregate patch predictions into slide-level classification, two methods were compared: majority voting and mean pooling. Moreover, two-stage aggregation was implemented to prioritize cancer tissues in slides.

In [20], scholars proposed three steps to develop an AI-based screening method for lymph node metastases. First, they trained a segmentation model to obtain lymph node tissue from WSI and broke it into patches. Next, they used a fine-tuned Xception model to classify patches into metastasis-positive/negative. Finally, the absence or presence of two connected patches classified as positive determined the final result of WSI. In [21], the authors compared the accuracies of stand-alone VGG-16 and VGG-19 models with ensemble models consisting of both architectures in classifying breast cancer histopathological images as carcinoma and non-carcinoma. In [22], the authors compared the performance of the VGG19 architecture with methods used in supervised learning with weakly labeled data to classify ovarian carcinoma histotype. The problem of binary classification into benign and malignant lesions, with subsequent division into eight subtypes with modified EfficientNetV2 architecture on images from the BreakHis dataset, was addressed by the authors in [23]. Similarly, Xception was employed in [24] for subtyping breast cancer into four categories. The binary subtype classification of eyelid carcinoma was performed in [25]. They used DenseNet-161 to make predictions for every patch in WSI and then used a patch voting strategy to decide the WSI subtype. In [26], the authors used AlexNet [27], GoogLeNet [28], and VGG-16 to detect histopathology images with cancer cells and to classify ovarian cancer grade. Since neural networks behave like black-box models, the authors employed the Grad-CAM method to demonstrate that CNN models attended to the cancer cell organization patterns when differentiating histopathology tumor images of different grades. Grad-CAM was also employed in [29], where the authors used this method to provide interpretability and approximate visual diagnosis for the presentation of the model's results to pathologists. The model consisted of three neural networks fine-tuned on a custom dataset to classify H&E stained tissue patches into five types of liver lesions, cirrhosis, and nearly normal tissue. A decision algorithm consisting of three networks was also proposed in [30] to detect odontogenic cyst recurrence using binary classifiers. The procedure consisted of letting the first two models make predictions. If the predictions did not match, a third model was loaded to obtain the final decision. Another example of using Grad-CAM is [31] to visualize classification results of the VGG16 network in grading bladder non-invasive carcinoma.

Hematoxylin-eosin (H&E) is considered as the gold standard for evaluating many cancer types. However, it contains only basic morphological information. In clinical practice, to obtain molecular information, immunohistochemical (IHC) staining is often employed. Such staining can visualize the expressions of different proteins (e.g., Ki67) on the cell membrane or nucleus. This approach is referred to as double staining. Many recent studies have shown that there is a correlation between H&E and IHC staining [32–34].

In [35], the authors addressed the problem of double staining in determining the number of Ki67-positive cells for cancer treatment. They employed matching pairs of IHC- and H&E-stained images and fine-tuned ResNet-18 at the cell-level from H&E images. Subsequently, to create a heat map, they transformed the CNN into a fully convolutional network without fully connected layers. As a result, the fine-tuned ResNet-18 was able to handle WSI as input and produce a heat map as output.

In [36], the authors proposed a modified Xception network called HE-HER2Net by adding global average pooling, batch normalization layers, dropout layers, and dense layers with a Swish activation function. The network was designed to classify H&E images into four categories based on Human epidermal growth factor receptor 2 (HER2) positivity from 0 to 3+. In addition to routine model evaluation, the authors compared their modified network to other existing architectures and claimed that HE-HER2Net surpassed all existing models in terms of accuracy, precision, recall, and AUC score.



To produce accurate models capable of generalization, it is essential to obtain large amounts of diversified data. Typically, this problem is addressed by pooling all necessary data to a centralized location. However, due to the nature of medical data, this approach has many obstacles regarding privacy and data ownership, as well as various regulatory policies (e.g., the General Data Protection Regulation GDPR of the European Union [37]). The authors of [38] simulated a Federated Learning (FL) environment to train a deep learning model that classifies cells and nuclei to identify TILs in WSI. They generated a dataset from WSIs of cancer from 12 anatomical sites and partitioned it into eight different nodes. To evaluate the performance of FL, they also trained a CNN using a centralized approach and compared the results. The study shows that the FL approach achieves similar performance to the model trained with data pooled at a centralized location.

**Table 1.** Summary of fine-tuning papers.

| Reference | Cancer Types | Staining | Dataset | Neural Networks in Models | Method |
|---|---|---|---|---|---|
| Abousamra et al. (2022) [14] | 23 cancer types | H&E | The Cancer Genome Atlas (TCGA) | Vgg-16, ResNet-34, InceptionV4 | Patch-level classification of Tumor infiltrating lymphocytes (TIL) |
| Yang et al. (2021) [18] | Lung cancer | H&E | Custom dataset of 1271 WSIs and 422 WSIs from TCGA | ResNet-50, EfficientNet-B5 | Six-type classification of lung lesions including pulmonary tuberculosis and Organizing pneumonia |
| Hameed et al. (2020) [21] | Breast cancer | H&E | Custom dataset of 544 WSIs | VGG-16, VGG-19 | Ensemble of neural networks to classify carcinoma and non-carcinoma images |
| Yu et.al (2020) [26] | Ovarian cancer | H&E | TCGA | AlexNet, GoogLeNet, VGG-16 | Cancerous regions identification and grades classification |
| Liu et al. (2020) [35] | Different types of cancer | H&E, IHC (Ki67) | Custom dataset from 300 Regions of interest | ResNet-18 | Classification of Ki67 positive and negative cells |
| Baid et al. (2022) [38] | 12 types | H&E | TCGA | VGG-16 | Federated learning for classification of tumor infiltrating lymphocytes |
| Cheng et al. (2022) [29] | Liver cancer | H&E | Custom dataset | ResNet50, InceptionV3, Xception | Ensemble of 3 networks pretrained on ImageNet used to differentiate Hepatocellular nodular lesions (5 types) with nodular cirrhosis and nearly normal liver tissue |
| Shovon et al. (2022) [36] | Breast cancer | H&E | BCI dataset | Modified Xception | Four class classification of HER2 with modified Xception model pretrained on ImageNet |
| Rao et al. (2022) [30] | Odontogenic cysts | H&E | Custom dataset | Inception-V3, DenseNet-121, Inception-Resnet-V2 | Binary classification of cyst recurrence based on decision algorithm consisting of 3 models |
| Farahani et al. (2022) [22] | Ovarian cancer | H&E | Custom dataset | VGG19 | Comparison of classification of ovarian carcinoma histotype by four models |
| Sarker et al. (2023) [23] | Breast cancer | H&E | BreakHis dataset | Modified EfficientNetV2 | Binary classification of malignant and benign tissue and multi-class subtyping using fused mobile inverted bottleneck convolutions and mobile inverted bottleneck convolutions with dual squeeze and excitation network and EfficientNetV2 as backbone |
| Luo et al. (2022) [25] | Eyelid carcinoma | H&E | Custom dataset | DenseNet161 | The differential diagnosis of eyelid basal cell carcinoma and sebaceous carcinoma based on patch prediction by the DenseNet161 architecture and WSI differentiation by an average-probability strategy-based integration module |
| Mundhada et al. (2023) [31] | Bladder cancer | H&E | Custom dataset | VGG16 | Grading of non-invasive carcinoma |
| Khan et al. (2023) [20] | Breast and colon cancer | H&E | PatchCamelyon | Xception | Segmentation of lymph node tissue with subsequent classification to detect metastases |
| Hameed et al. (2022) [24] | Breast cancer | H&E | Colsanitas dataset | Xception | Using Xception networks as feature extractor to classify breast cancer into four categories: normal tissue, benign lesion, in situ carcinoma, and invasive carcinoma |

### 3.3. Training from Scratch

As already stated, fine-tuning is a promising method for training deep neural networks. On the other hand, it can only be applied to well-known architectures that are already pre-trained. When designing a custom CNN architecture, it needs to be trained from scratch. Table 2 summarizes studies in which neural networks were trained from scratch. In [39], the authors proposed a method based on CNN with residual blocks (Res-Net) referred to as DeepLRHE to predict lung cancer recurrence and the risk of metastasis. Later in [40], scholars established the new DeepIMHL model consisting of CNN and Res-Net to predict mutated genes as biomarkers for targeted-drug therapy of lung cancer. In addition, the authors in [41] trained and optimized EfficientNet models on images of non-Hodgkin lymphoma and evaluated its potential to classify tumor-free reference lymph nodes, nodal small lymphocytic lymphoma/chronic lymphocytic leukemia, and nodal diffuse large B-cell lymphoma. In [42], the authors proposed three architectures of ResNet differing in the construction of residual blocks trained from scratch. Their suggested model achieved accuracy comparable to other state-of-the-art approaches in the classification of oral cancer histological images into three stages. To classify kidney cancer subtypes, in [43] the authors developed an ensemble-pyramidal model consisting of three CNNs that process images of different sizes. The authors in [44] demonstrated that CNN-based DL can predict the gBRCA mutation status from H&E-stained WSIs in breast cancer. According to researchers in [45], CNN can be employed to differentiate non-squamous Non-Small Cell Lung Cancer versus squamous cell carcinoma. To classify the tumor slide, they pooled information using the max-pooling strategy. Moreover, they added quality check with a threshold for predictions to select only tiles with a high prediction level. Additionally, to improve the prediction, they also used a virtual tissue microarray (circle from the centroid based on the pathologist's hand-drawn tumor annotations) instead of WSI.

To compare the performance of pre-trained networks with the custom ones trained from scratch, researchers in [46] used images of three cancer types: melanoma, breast cancer, and neuroblastoma. Unlike others using patches, the authors applied the simple linear iterative clustering (SLIC) to segment images into superpixels which group together similar neighboring pixels, as shown in Figure 4. Thus, these superpixels were classified into multiple subtype categories based on the type of cancer. To make WSI-level predictions, they used multiple specific quantification metrics such as stroma-to-tumor ratio. Although the custom NN achieved comparable results, pre-trained networks performed better on all three cancer types. A similar comparison was carried out in [47] for the classification of subtypes in lung cancer biopsy slides. Results showed that a CNN model built from scratch fitted to the specific pathological task could produce better performances than fine-tuning pre-trained CNNs.

A comparison of training from scratch versus transfer learning was performed in [48]. The authors compared three approaches for training the VGG16 network: training from scratch, transfer learning as a feature extractor, and fine-tuning on images of breast cancer to detect Invasive Ductal Carcinoma. According to the results, the model trained from scratch achieved better results in terms of accuracy (0.85). However, using transfer learning, they were able to train a comparable model (accuracy 0.81) ten times faster. Furthermore, among the transfer learning approaches, transfer learning via feature extraction (accuracy 0.81), which involved retraining some of the convolutional blocks, yielded better results in less time compared to transfer learning via fine-tuning (accuracy 0.51).

**Table 2.** Summary of papers training neural networks from scratch.

| Reference | Cancer Types | Staining | Dataset | Neural Networks in Models | Method |
|---|---|---|---|---|---|
| Wu et al. (2020) [39] | Lung cancer | H&E | 211 samples from TCGA | Custom CNN with residual blocks | Prediction of lung cancer recurrence |
| Huang et al. (2021) [40] | Lung cancer | H&E | TCGA | Custom CNN with residual blocks | Identification of the bio-markers of lung cancer |
| Steinbuss et al. (2021) [41] | Blood cancer | H&E | Custom dataset from 629 patients | EfficientNet | Classification of tumor-free lymph nodes, nodal small lymphocytic lymphoma/chronic lymphocytic leukemia, and nodal diffuse large B-cell lymphoma |
| Panigrahi et al. (2022) [42] | Oral cancer | H&E | Custom dataset | Three ResNet architectures | Classification of 3 grades |
| Wang et al. (2021) [44] | Breast cancer | H&E | Custom dataset of 222 images | ResNet-18 | BRCA gene mutations prediction |
| Le Page et al. (2021) [45] | Lung cancer | H&E | Custom dataset of 197 images and 60 images from TCGA | InceptionV3 | Classification of patches (tiles) into cancer subtypes. For final case classification they used majority-vote method or highest probability class |
| Zormpas-Petridis et al. (2021) [46] | Melanoma, breast cancer and childhood neuroblastoma | H&E | Custom dataset | Custom CNN | Classification of the: melanoma (tumor tissue, stroma, cluster of lymphocytes, normal epidermis, fat, and empty/white space) breast cancer (tumor, necrosis, stroma, cluster of lymphocytes, fat, and lumen/empty space) neuroblastoma (undifferentiated neuroblasts, tissue damage (necrosis/apoptosis), areas of differentiation, cluster of lymphocytes, hemorrhage, muscle, kidney, and empty/white space) |
| Abdolahi et al. (2020) [48] | Breast cancer | H&E | Kaggle | Custom CNN, VGG-16 | Classification of invasive ductal carcinoma |
| Yang et al. (2022) [47] | Lung cancer | H&E | Custom dataset | Custom CNN | Comparison of classification lung cancer by fine-tuned models and models trained from scratch |
| Abdeltawab et al. (2022) [43] | Kidney cancer | H&E | Custom dataset | Custom CNN | An ensemble-pyramidal deep learning model consisting of three CNNs processing different image sizes to differentiate 4 tissue subtypes |

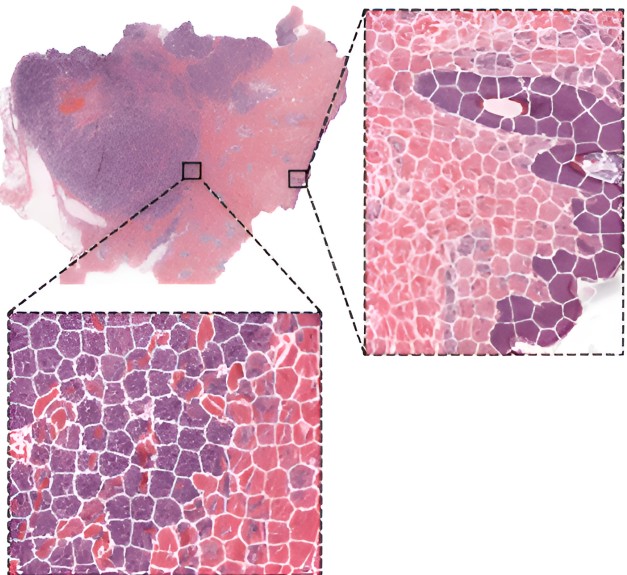

**Figure 4.** WSI image segmentation using the SLIC superpixels algorithm. Reprinted from [46], with permission according to Creative Commons Attribution License.

### 3.4. Multi-Stage Classification

In [49], scholars tackled the complex problem of computer-aided disease diagnosis by designing a two-stage system to determine the Tumor Mutation Burden (TMB) status, which is an important biomarker for predicting the response to immunotherapy in lung cancer. For the first stage, they developed a CNN based on InceptionV3 [50] to classify known histologic features for individual patches across H&E-stained WSIs. In the second stage, the patch-level CNN predictions were aggregated over the entire slide and combined

with clinical features such as smoking status, age, stage, and sex to classify the TMB status. The final model was obtained by ensembling 10 independently trained networks.

In [51], the authors proposed a diagnostic framework for generating a whole-case report consisting of the detection of renal cancer regions, classification of cancer subtypes, and cancer grades. From every stain-normalized WSI, patches were selected from tumor and non-tumor regions to form a dataset. For tumor region classification, they fine-tuned several different architectures and identified InceptionV3 as the most suitable one. Thus, they also used this architecture for the remaining tasks. Patches classified as containing a tumor were further classified into three tumor subtypes and four grade classes.

It should be noted that CNNs have proven to be successful classifiers in the field of histology. Nevertheless, they can also be employed in conjunction with other machine learning (ML) classifiers. The authors in [52] developed a CNN model for the automated classification of pathology glioma (brain tumor) images into six subtypes. The images pass through the CNN to obtain patch-level output categories. At this point, those patch labels go through a hierarchical decision tree for patient-level diagnosis based on the amounts and proportions of tumor types. The outcome thus includes results for both the image patch-label and the patient-level label. In [53], researchers developed a three-step approach to HER2 status tissue classification in breast cancer. Firstly, they used a pre-trained UNet-based nucleus detector [54] to create patches. Secondly, they trained a CNN to identify tumor nuclei and further classified them as HER2-positive or HER2-negative. In [55], the authors proposed a classification method for subtype differentiation of liver cancer based on a stacking classifier with deep neural networks as feature extractors. They used four pre-trained deep convolutional neural networks, ResNet50, VGG16, DenseNet201 [56], and InceptionResNetV2 [17], to extract deep features from histopathological images. After fusing extracted deep features from different architectures, they applied multiple ML classifiers (Support-vector machines (SVMs), k-Nearest Neighbor (k-NN), Random Forest (RF)) on the feature vector to obtain final classification.

To predict 5-year overall survival in renal cell carcinoma, scholars in [57] fine-tuned a ResNet18 pre-trained on the ImageNet dataset. The CNN assigned a probability score for every patch, and to determine the class for an entire WSI, the scores of all associated patches were averaged and classified. In addition to single-stage classification, they also used CNN prediction with other clinicopathological variables for multivariable logistic regression analysis. The authors in [58] presented an approach that combines a deep convolutional neural network as a patch-level classifier and XGBoost [59] as a WSI-level classifier to automatically classify H&E-stained breast digital pathology images into four classes: normal tissue, benign lesion, ductal carcinoma in situ, and invasive carcinoma. InceptionV3 was trained as the Patch-Level Classifier to generate four predicted probability values combined into a heatmap. By comparing the classification accuracy of different classifiers, they chose XGBoost as the WSI-level classifier.

In [60], researchers trained a deep learning classifier and applied it to classify lung tumor samples into nine tissue classes. From the extracted features, they computed spatial features that describe the composition of the tumor microenvironment and used them in combination with clinical data to predict patient survival, as well as to predict tumor mutation. The authors of [61] claim that they were the first to propose a method for detecting Pancreatic ductal adenocarcinoma in WSIs based on CNNs. They employed InceptionV3 as a patch-level classifier and predicted patches combined with a malignancy probability heatmap. At this point, statistical features were extracted from WSI heatmaps and applied to train a Light Gradient Boosting Machine [62] for slide-level classification. Similar approaches were taken by researchers in [63]. On histological images of gastric cancer (GC), they made both binary and multi-class classifications. Firstly, InceptionV3 was used for both malignant and benign patch classification as well as discriminating normal mucosa, gastritis, and gastric cancer. Secondly, they separated all WSIs into categories, "complete normal WSIs" and "mixture WSIs" with gastritis or GC, and used 44 features

extracted from the malignancy probability heatmap generated by CNN to train and fine-tune the RF classifier.

The addition of attention mechanisms to CNNs for increased performance has become increasingly popular nowadays. In [64], the Divide-and-Attention Network (DANet) was proposed for breast cancer classification and grading of both breast and colorectal cancers. This network has three inputs: the original pathological image, the nuclei image, and the non-nuclei image. The nuclei and non-nuclei images are obtained as a result of a nuclei segmentation model. A similar approach was used in [65], where the authors developed the Nuclei-Guided Network (NGNet) for grading of breast invasive ductal carcinoma. Compared to DANet, NGNet has only two input images: the original image and the nuclei image obtained from segmentation.

Medulloblastoma (MB) is a dangerous malignant pediatric brain tumor that can lead to death [66]. In [67], the authors proposed a mixture of deep learning and machine learning methods called MB-AI-His for the automatic diagnosis and classification of four subtypes of pediatric MB. The diagnosis is performed in two levels. The first level classifies the images into normal and abnormal (binary classification level), while the second level classifies the abnormal images containing MB tumor into the four subtypes of childhood MB tumor (multi-classification level). Three pre-trained deep CNNs are utilized with transfer learning (ResNet-50, DenseNet-201, and MobileNet [68]) to extract spatial features. These features are combined with time-frequency features extracted using the discrete wavelet transform (DWT) method. Finally, a combination of spatial features and five popular classifiers is used to perform multi-class classification, including SVM, k-NN, Linear Discriminant Analysis, and Ensemble Subspace Discriminant. A similar approach is introduced by the authors in [69]. Multi-class classification of the four classes of childhood MB is much more complicated than binary classification. Few research articles have investigated this multi-class classification problem. Their pipeline consists of spatial DL feature extraction from 10 fine-tuned CNN architectures, feature fusion and reduction using the DWT method, and subsequent selection of features. Classification is accomplished using a bidirectional Long-Short-Term Memory classifier. All papers using multistage classification are listed in Table 3.

**Table 3.** Summary of studies using multi-stage classification.

| Reference | Cancer Types | Staining | Dataset | Neural Networks in Models | Method |
|---|---|---|---|---|---|
| Sadhwani et al. (2021) [49] | Lung cancer | H&E | TCGA and custom dataset of 50 WSIs | Custom CNN | Multiclassification into subtypes and binary classification of Tumor Mutation Burden |
| Wu et al. (2021) [51] | Renal cell cancer (RCC) | H&E | 667 WSIs from TCGA + new RCC dataset of 632 WSIs | InceptionV3 | Identification of tumor regions and classification into tumor subtypes and different grades |
| Jin et.al (2021) [52] | Brain cancer | H&E | slides of 323 patients from the Central Nervous System Disease Biobank | custom CNN based on DenseNet | Classification into 5 subtypes of glioma |
| Anand et al. (2020) [53] | Breast cancer | H&E, IHC | dataset from University of Warwick and TCGA | Custom neural network | Identification of tumor patches and classification of HER2 into positive or negative |
| Dong et al. (2022) [55] | Liver cancer | H&E | Custom dataset of 73 images | ResNet-50, VGG-16, DenseNet-201, InceptionResNetV2 | Classification of three differentiation states |
| Mi et al. (2021) [58] | Breast cancer | H&E | Custom dataset of 540 WSIs | InceptionV3 | Multi-class classification of normal tissue, benign lesion, ductal carcinoma in situ, and invasive carcinoma |
| Fu et al. (2021) [61] | Pancreas | H&E | Custom dataset of 231 WSIs | InceptionV3 | Classification of patches into cancerous or normal |
| Ma et al. (2020) [63] | Gastric cancer | H&E | Custom dataset of 763 WSIs | InceptionV3 | Classification of normal mucosa, chronic gastritis, and intestinal-type |

**Table 3.** *Cont.*

| | | | | | |
|---|---|---|---|---|---|
| Attallah (2021) [67] | Brain cancer | H&E | Custom dataset of 204 images | ResNet-50, DenseNet-201, MobileNet | Classification of normal and abnormal Medulloblastoma |
| Attallah (2021) [69] | Brain cancer | H&E | Custom dataset of 204 images | 10 CNN architectures | Multi-class classification of 4 medulloblastoma subtypes |
| Yan et al. (2022) [64] | Breast and colorectal cancer | H&E | BACH dataset and datasets avaiable from different articles | Xception | Classification of breast cancer, colorectal and breast cancer grading based on Divide-and-Attention Network using Xception CNN as backbone |
| Yan et al. (2022) [65] | Breast cancer | H&E | Custom dataset | NGNet | Grading of breast cancer using attention modules and segmentation. Classification is done with two images: original image and corresponding nuclei image) |
| Raczkowski et al. (2022) [60] | Lung cancer | H&E | Custom dataset | ARA-CNN | Classification of mutation based on tissue prevalence and tumor microenvironment composition computed from ARA-CNN output. CNN was used to classify patches into 9 tissue subtypes |
| Wessels et al. (2022) [57] | Kidney cancer | H&E | TCGA | ResNet18 | Pretrained ResNet18 CNN was used to predict 5-year overal survival in renal cell carcinoma. Furthermore, the CNN-based classification was an independent predictor in a multivariable clinicopathological model |

## 4. Discussion

Based on the studies described in the previous chapter, it is clear that there are many approaches to successfully use neural networks for many classification tasks in histology and a variety of cancer types. Most commonly, DL has been applied to lung and breast cancer. Breast cancer is a leading cause of cancer-related deaths in women worldwide, and lung cancer was the second most commonly diagnosed cancer worldwide in 2020, behind female breast cancer [24,47]. From a histological point of view, the tasks in which neural networks were successfully applied have been divided into the following three groups: tissue types, grading, and biomarker classification. The articles mentioned in this review are arranged according to this categorization in Table 4.

**Table 4.** Overview of all studies classified according to the application area.

| | | | |
|---|---|---|---|
| Tissue | Tissue type | Yang et al. (2021) [18]<br>Hameed et al. (2020) [21]<br>Farahani et al. (2022) [22]<br>Luo et al. (2022) [25]<br>Hameed et al. (2022) [24]<br>Le Page et al. (2021) [45]<br>Abdolahi et al. (2020) [48]<br>Abdeltawab et al. (2022) [43]<br>Wu et al. (2021) [51]<br>Anand et al. (2020) [53]<br>Mi et al. (2021) [58]<br>Ma et al. (2020) [63]<br>Yan et al. (2022) [64] | Yu et.al (2020) [26]<br>Cheng et al. (2022) [29]<br>Sarker et al. (2023) [23]<br>Khan et al. (2023) [20]<br>Steinbuss et al. (2021) [41]<br>Zormpas-Petridis et al. (2021) [46]<br>Yang et al. (2022) [47]<br>Sadhwani et al. (2021) [49]<br>Jin et.al (2021) [52]<br>Dong et al. (2022) [55]<br>Fu et al. (2021) [61]<br>Attallah (2021) [69]<br>Attallah (2021) [67] |
| | Tissue grading | Yu et.al (2020) [26]<br>Wu et al. (2021) [51]<br>Panigrahi et al. (2022) [42] | Mundhada et al. (2023) [31]<br>Yan et al. (2022) [65] |
| Biomarkers | Abousamra et al. (2022) [14]<br>Shovon et al. (2022) [36]<br>Anand et al. (2020) [53] | Liu et al. (2020) [35]<br>Huang et al. (2021) [40]<br>Raczkowski et al. (2022) [60] | Baid et al. (2022) [38]<br>Wang et al. (2021) [44] |

### 4.1. Tissue Types

One of the most fundamental tasks in histology is the classification of tissue types. It is possible to look at this task in two ways. The first aspect and the complete basis is to identify tumor tissue and other tissue types. This may involve a binary division into tumor and non-tumor tissue (this approach was used in [21]) as well as multi-class detection of tumor, stroma, lymphocytes, fat, necrosis, and other. In [46], the authors demonstrated that their proposed SuperHistopath framework succeeded in tissue multi-classification of three

different cancer types and was able to achieve high accuracy (98.8% in melanomas, 93.1% in breast cancer, and 98.3% in childhood neuroblastoma).

The second aspect is classifying tumor tissue into cancer subtypes. This could be the classification of malign vs. benign carcinoma, invasive vs. non-invasive carcinoma, or various subtypes of a certain cancer type. This subtyping is an important part of determining a treatment plan; however, it often needs special IHC staining to be done. Therefore, the ability to perform subtype classification directly from H&E images could be of great benefit in terms of clinical application. Authors in [23] proposed a method for subdividing breast cancer into eight subtypes, four for benign (adenosis, fibroadenoma, phyllodes tumour, and tubular adenoma) and four for malignant (carcinoma, lobular carcinoma, mucinous carcinoma, and papillary carcinoma). They showed that their model achieved significant results compared to other state-of-the-art models mentioned in the study.

It should be noted that the two approaches are not always clearly separable, and the classification of tissue type is often associated with the classification of tumor subtypes. This approach was demonstrated in [24], where the breast tissue was categorized as normal tissue, benign lesion, in situ carcinoma, or invasive carcinoma. Another example is [29], where researchers managed to obtain models with accuracy over 0.95% in classifying five types of liver lesions, cirrhosis, and nearly normal tissue.

### 4.2. Tissue Grading

Cancer grading has its origins in 1914 when pathologist Albert Broders began collecting data showing that cancers of the same histologic type behaved differently. By the late 1930s, tumor grading was considered a state-of-the-art prognostic technique for scientific cancer care. Today, there are hundreds of grading schemes for various types of cancer [70]. However, in comparison with subtype classification, pathological image grading is considered a fine-grained task [64,65].

Researchers in [42] used residual networks to grade images of squamous cell carcinoma, since it accounts for about 90% of oral disorders. To demonstrate the deep learning capability of grading different cancer types, the authors of [64] developed a model with an average classification accuracy of 95% and 91% for colorectal and breast cancer grading, respectively. The breast cancer grading task was also addressed in [65].

### 4.3. Bio-Marker Classification

A bio-marker is a biological molecule found in tissues that is a sign of a normal or abnormal process or of a condition or disease, such as cancer. Typically, bio-markers differentiate a person without disease from an affected patient. There is a tremendous variety of bio-markers, including proteins, antibodies, nucleic acids, gene expression, and others. They can be used in clinical treatment for multiple tasks, such as estimating the risk of disease, differential diagnosis, predicting response to therapy, determining the prognosis of the disease, and so on [71].

In [36], the authors presented the architecture HE-HER2Net, which surpassed the accuracy of other common architectures in the multiclassification of HER2 into four categories. Following this, researchers in [40] developed a CNN to predict the mutated genes, which are potential candidates for targeted-drug therapy for lung cancer. The average probability of the bio-markers of lung cancer was received through the model, with the highest accuracy of 86.3%.

Ki67 is a protein that is found in the cancer cell nucleus and can be found only in cells that are actively growing and dividing, which is typical for cells mutated into cancer. Therefore, Ki67 is sometimes considered a good marker of proliferation (rapid increase in the number of cells) [72]. In [35], scholars fine-tuned an NN to classify cell images into Ki67-positive, Ki67-negative, and as a background image with an accuracy of 93%.

## 5. Conclusions

The article presents a detailed survey of recent DL models based on neural networks in the context of classification tasks for the analysis of histological images. The analysis of approximately 70 articles published in the last three years shows that automated processing and classification of histopathological images by deep learning methods have been applied to a wide range of histological tasks, such as tumor tissue classification or biomarker evaluation to determine treatment plans. The survey reveals several conclusions:

**Application Areas:** Deep learning has been applied to several types of cancer (e.g., breast, lung, colon, brain, kidney) and has proven to be capable of assisting pathologists with visual tasks in the treatment of various diseases. The reviewed works have identified the following three groups of specific tasks: classification of tissue type, grading of specific tissue, and identification of the presence of biomarkers.

**Single- and Multi-Stage Approaches:** Convolutional neural networks can be applied either as a stand-alone classifier or can be used as a feature extractor whose outputs will proceed into another machine learning model to carry out the final classification.

**Pre-Training:** Training networks from scratch requires a large dataset and a lot of computing time. Therefore, it is recommended to experiment with well-known architectures pre-trained on ImageNet. If the results are not sufficient, then one can design their own custom network and train it from scratch.

**Author Contributions:** Conceptualization, D.P. and I.C.; resources, D.P.; writing—original draft preparation, D.P.; writing—review and editing, D.P. and I.C.; visualization, D.P.; supervision, I.C.; funding acquisition, I.C. All authors have read and agreed to the published version of the manuscript.

**Funding:** This research was supported by the Operational Program "Integrated Infrastructure" of the project "Integrated strategy in the development of personalized medicine of selected malignant tumor diseases and its impact on life quality", ITMS code: 313011V446, co-financed by resources of European Regional Development Fund.

**Data Availability Statement:** Not applicable

**Conflicts of Interest:** The authors declare no conflict of interest.

## Abbreviations

The following abbreviations are used in this manuscript:

| | |
|---|---|
| AI | Artificial Intelligence |
| AUC | Area Under Curve |
| BCI | Breast Cancer Immunohistochemical |
| DL | Deep Learning |
| CNNs | Convolutional Neural Networks |
| WSIs | Whole Slide Images |
| NNs | Neural Networks |
| TIL | Tumor Infiltrating Lymphocytes |
| H&E | Hematoxylin and Eosin |
| IHC | Immunohistochemical |
| FL | Federated Learning |
| GDPR | General Data Protection Regulation |
| SLIC | Simple Linear Iterative Clustering |
| TMB | Tumor Mutation Burden |
| ML | Machine Learning |
| HER2 | Human Epidermal Growth Factor Receptor 2 |
| SVM | Support-vector machines |
| k-NN | k-Nearest Neighbor |
| RF | Random Forest |
| GC | Gastric Cancer |

| | |
|---|---|
| MB | Medulloblastoma |
| DWT | Discrete Wavelet Transform |
| RCC | Renal cell cancer |

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
