# Peer review of "Survey of Recent Deep Neural Networks with Strong Annotated Supervision in Histopathology"

_computation, doi:10.3390/computation11040081_

Round 1
Reviewer 1 Report
The overall manuscript is acceptable, but it needs some improvement for better understanding for readers. The author needs to address the following issues
· The novelty of the manuscript should be improved in the revised version.
· The author must write separate section discussion sections in the revised version.
· In the discussion, section author includes significant fruitful results. The compression of state-of-the-art works with graphical representation models of the revised version.
· The conclusion section was very weak; it needs to be improved and added to the scope of feature work and research gap identified by the author.
· There are many punctuation/grammatical mistakes that should be removed.
Reviewer 2 Report
The manuscript presents a overview in the usage of Deep Neural Networks in the classification of histopatology images.
Authors found that automated processing and classification of histopathological images by deep learning methods was addressed to a wide range of histological tasks such as tumor tissue classification or biomarker evaluation to determine treatment plan, but it has possibilities of been applied further.
I find the topic interesting and being worth of investigation and the document is well structured.
Although I propose the following comments/suggestions:
- Abstract should be better organized: problem, motivation, aim, methodology, main results, further impact of those results.
- Keywords should be in alphabetical order
- I strongly suggest authors from refraining using personal pronouns such as "we" and "our" throughout the text and I encourage them to write it in an impersonal form of writing.
- In terms of methodology the manuscript misses repeatability by not disclosing the literature sources and keywords and neither the inclusion and exclusion criteria.
- It would be recommended to use a PRISMA methodology.
- The conclusions are vague and biased.
Reviewer 3 Report
This review paper can be accepted after revision based on the following comments.
1. At present, there are many methods based on Transformer in histopathology, which are different from neural network and CNN. If the authors can cover this aspect, the article will be even better.
2. If the authors can show the relationship between the mentioned methods in the form of atlas, readers can intuitively feel the application of deep learning in histopathology.
3. If deep learning can be applied to histopathology, it will be better to classify it in more detail from the perspective of tasks.
4. If the authors can list the performance evaluation indicators in the paper summary table, readers will have an intuitive understanding of the advantages and disadvantages of the existing methods.
Round 2
Reviewer 1 Report
The author addressed all the comments in the revised manuscript. So, I accept it in its present form.
Author Response
Dear reviewer,
thank you for your conclusion.
Reviewer 2 Report
The authors have addressed the comments and suggestions, improving the manuscript, therefore I am favourable for it to be accepted as is.
Author Response

(The authors gave the same response as above.)
